# Collective total synthesis of stereoisomeric yohimbine alkaloids

Meiyi Tang[1,2], Haigen Lu[1,2] & Liansuo Zu ®[1] ✉

Stereoisomeric polycyclic natural products are important for drug discovery-based screening campaigns, due to the close correlation of stereochemistry with diversified bioactivities. Nature generates the stereoisomeric yohimbine alkaloids using bioavailable monoterpene secologanin as the ten-carbon building block. In this work, we reset the stage by the development of a bioinspired coupling, in which the rapid construction of the entire pentacyclic skeleton and the complete control of all five stereogenic centers are achieved through enantioselective kinetic resolution of an achiral, easily accessible synthetic surrogate. The stereochemical diversification from a common intermediate allows for the divergent and collective synthesis of all four stereoisomeric subfamilies of yohimbine alkaloids through orchestrated tackling of thermodynamic and kinetic preference.

Natural products have long been a rich source for the discovery of novel therapeutics as exemplified by the developments of numerous drugs thereof[1]. In modern drug discovery-based screening campaigns, stereochemical diversity of the compound libraries has been recognized as one of the key factors for success[2]. Merging these two aspects emphasizes the value of stereoisomeric polycyclic natural molecules, which feature stereochemical divergence on the same molecular scaffold. Among molecules in this category, the secologanin-derived alkaloid skeletons (e.g., **I–IV**, Fig. 1A)[3–5] are noteworthy, which constitute a large compound collection and demonstrate high level of stereochemical diversity. While collective total synthesis of related indole alkaloids with structural diversity has been reported[6,7], the chemical assembly of compound collections based on stereoisomeric skeletons, particularly those with multiple stereochemical variations, has been less common. Secologanin (**6**; Fig. 1C) is widely used by nature as a 10-carbon building block to generate more than 3000 monoterpene alkaloids, however, biomimetic approaches are hampered by its limited commercial availability (5 mg, >200$, Aldrich) or no-so-easy accessibility by asymmetric chemical synthesis[8–10]. We surmised to reset the stage by the development of a bioinspired coupling, in which the chiral secologaninin-type building block was replaced by an achiral, easily accessible synthetic surrogate and the stereochemistry was set up through enantioselective kinetic resolution. In addition, stereochemical diversification of one stereoisomeric

skeleton to others should be preferred over the individual coupling of multiple stereochemical-defined fragments. Herein, we demonstrate the synthetic power of such concepts, namely bioinspired coupling and stereochemical diversification, using yohimbine alkaloids as the proving grounds.

The yohimbine alkaloids are historically renowned, and structurally demonstrate high degrees of stereochemical variations. As exemplified by the structures of five representative members (Fig. 1B), yohimbine (**1a**), β-yohimbine (**1b**), pseudoyohimbine (**2**), alstovenine (**3**) and venenatine (**4**), share the same pentacyclic framework (ABCDE, skeleton **I**, Fig. 1A), but differ in the stereochemical arrangement of four of their five stereogenic centers (C3, C20, C16, C17). The stereochemical difference at the ring junction (C3/C20) serves as the signature to divide these alkaloids into four subfamilies: *normal* (**1a, 1b**), *pseudo* (**2**), *allo* (**3**), *epiallo* (**4**). The changes in stereochemistry result in notable alternation of the three-dimensional conformation and spatial orientation of functional groups, and thus, are closely correlated to the diversified bioactivities. For example, alstovenine (**3**) and its C3-epimer venenatine (**4**) display contrasting activity on central nervous system (CNS) by enhancing or inhibiting the analgesic effects of morphine, respectively[11]. Of note, the approved uses of several members, such as yohimbine and reserpine, as therapy for human diseases, emphasize the significance of privileged yohimbine skeleton for

[1]School of Pharmaceutical Sciences, Key Laboratory of Bioorganic Phosphorus Chemistry & Chemical Biology (Ministry of Education), Beijing Frontier Research Center for Biological Structure, Tsinghua University, Beijing 100084, China. [2]These authors contributed equally: Meiyi Tang, Haigen Lu. ✉e-mail: zuliansuo@tsinghua.edu.cn

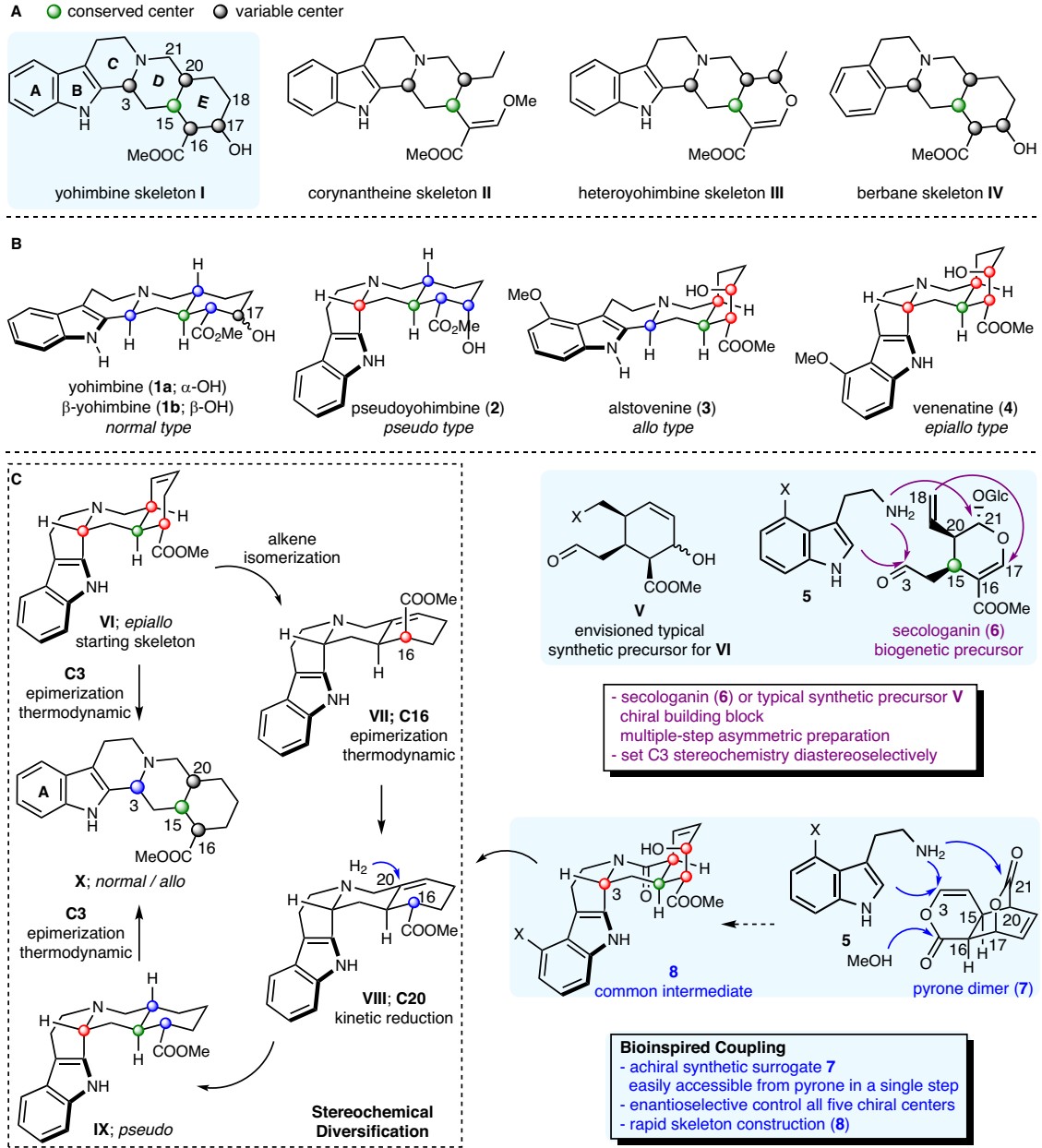

**Fig. 1 | General synthetic plan for stereoisomeric yohimbine alkaloids. A** secologanin-derived alkaloid skeletons with stereochemical divergence. **B**: representative stereoisomeric yohimbine alkaloids. **C**: stereochemical diversification and bioinspired coupling strategies.

drug discovery programs[12,13], particularly for CNS disorders, which historically have enjoyed the use of natural products as leads[14].

Since Woodward's landmark synthesis of reserpine (*epiallo* type) in 1956[15], various innovative synthetic approaches have been developed, leading to the chemical synthesis of the specific stereoisomer of this class[16–18]. While promising progress has been elegantly made by the groups of Stork, Sarpong and Scheidt in divergent addressing the C3-stereochemistry[19–21], the enantioselective, divergent and collective synthesis of all four stereoisomeric subfamilies of yohimbine alkaloids remains an unsolved problem. Thus, the high degrees of stereochemical information imbedded in these molecules could not be fully explored for biomedical applications. There are four stereochemical variations in the yohimbine skeletons (C3, C20, C16, C17), which could not be randomly adjusted bidirectional. A successful synthetic plan should dictate which stereoisomeric skeleton to be made at the outset, from which other skeletons could be derived through stereochemical diversification by following thermodynamic/kinetic preference.

Putting these concerns together cultivated a simplified plan (Fig. 1C; without considering the C17-OH), in which the *epiallo* skeleton **VI** was determined as the starting point due to the favorable C3-epimerization (to **X**) under thermodynamic control and possibility of using the alkene to direct stereochemistry adjustment at C16 (**VII** to **VIII**, thermodynamic epimerization) and C20 (**VIII** to **IX**, kinetic reduction). Thus, the rapid construction of an intermediate resembling **VI** was required. Typical synthetic precursor **V** could be envisioned, but its asymmetric synthesis should be not easy. Both biogenetic precursor secologanin (**6**, Fig. 1C) and the envisioned synthetic building block **V** are chiral, and would synthetically dictate the C3-stereochemistry by diastereoselective induction. We surmised a bioinspired coupling strategy, in which the sequence of stereochemistry generation was reversed and the tedious synthesis of a chiral precursor could be bypassed. Specifically, as depicted in Fig. 1C, we envisioned to use pyrone dimer **7** as an achiral 10-carbon building block, which can be made in a single step and bears similar functionality as that of

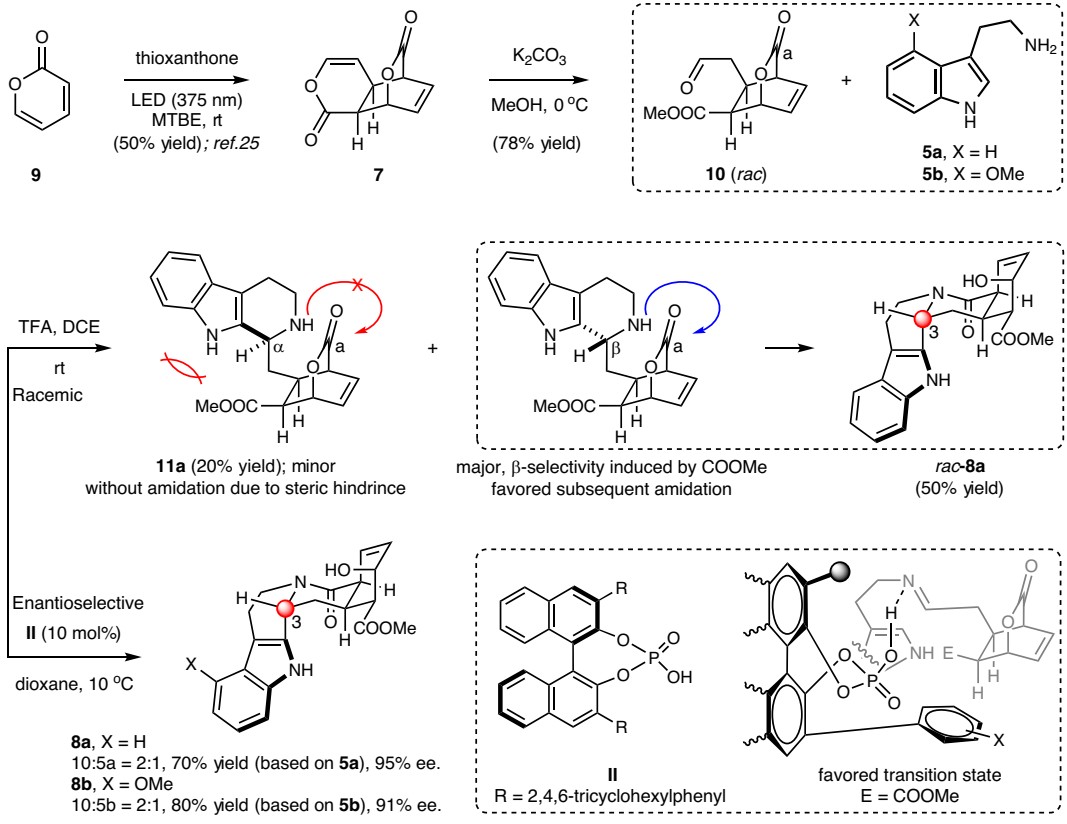

**Fig. 2 | Three-step construction of the pentacyclic skeleton.** Racemic version used trifluoroacetic acid (TFA) as the promotor. Enantioselective version used chiral phosphoric acid catalyst **II**. MTBE: methyl *tert*-butyl ether.

secologanin (**6**) (upon C17–C18 linkage). If the enantioselective Pictet-Spengler cyclization of tryptamine **5** and **7** (or its derivative) followed by selective amidation at C21 could occur, the entire yohimbine skeleton with requisite functionality (**8**) would be generated with the complete control of all five stereogenic centers through kinetic resolution. The realization of these concepts would collectively cultivate a general synthetic platform for the synthesis of stereoisomeric yohimbine natural products using only tryptamine and pyrone as the starting materials in an enantioselective, atom-efficient[22], step-efficient[23], protecting-group-free[24], concise and divergent manner.

## Results

### 3-Step construction of the pentacyclic skeleton
The known pyrone dimer **7** was prepared by following a previously established protocol[25] for the photo-dimerization of 2-pyrone **9** using thioxanthone (2 mol%) as the photo-sensitizer (Fig. 2). This single step dimerization significantly increases the structural complexity by doubling the number of atoms, incorporating two new chemical bonds, and converting the flat monomer to a three-dimensional (3D) dimeric skeleton with four stereogenic centers. Our initial attempts on direct harnessing the enol moiety of **7** for Pictet-Spengler cyclization with tryptamine **5a**, under a variety of acidic conditions, were not successful. Thus, **7** was ring-opened in the presence K₂CO₃ in methanol to release the aldehyde **10**, which turned out to be a suitable substrate for the Pictet-Spengler/amidation cascade. In a racemic synthesis using trifluoroacetic acid (TFA) as the acid, *rac*-**8a** was observed as the major product, accompanying by the C3-diastereomeric (C3-α) Pictet-Spengler product **11a** as the minor product without further amidation. These results indicated that the diastereoselectivity (C3) of the racemic Pictet-Spengler cyclization was about 2.5:1, and only the major diastereomer (C3-β) engaged in the subsequent amidation with COᵃ. The COOMe group played a critical role in directing the selectivities of

this transformation by dictating the facial selectivity of the Pictet-Spengler cyclization and preventing **11a** toward further amidation due to steric hindrance (Fig. 2). The stereochemistry of **8a** corresponds to the *epiallo* type yohimbine alkaloids (Fig. 1). The late stage setting up the desired C3-stereochemistry of this type has been a long-standing challenge; our method offered an alternative way to address this problem through kinetic amidation, which is mechanistically different from Stork/Sarpong's aminonitrile/Pictet-Spengler solution[19,20].

Based on these results, we turned to enantioselective synthesis of **8a** applying chiral phosphoric acid catalysis[26,27]. While catalytic asymmetric Pictet-Spengler cyclization promoted by this class of catalysts has been known[28–30], the application to direct synthesis of a functionalized product as complicated as **8a** has been rare. The process would be the kinetic resolution of racemic **10** in nature. While such an operation would inherently lead to the waste of half of **10**, the early-stage setting and the readily accessible of this substrate from simple 2-pyrone allowed us to use **10** in excess and use **5a** as the limiting reagent. The success of the catalytic process would hinge on the ability of the chiral catalyst in controlling the C3-β selectivity in an enantioselective manner with the matched half of **10** as envisioned by the depicted transition state (Fig. 2). The detailed optimization of reaction conditions was summarized in the supplementary information (Supplementary Table 1). The best results were obtained using catalyst **II** (10 mol%) in dioxane at 10 °C, affording **8a** in 70% yield and 95% ee. The absolute stereochemistry of **8a** was confirmed by X-ray crystallographic analysis. The use of **5b** as the substrate under identical conditions led to the synthesis of **8b** in 80% yield and 91% ee, indicating the tolerance of the structural variation of the tryptamine part. Overall, the 3-step enantioselective construction of the entire pentacyclic skeleton with requisite functionality provided us the common intermediates (**8**) for the total synthesis of yohimbine alkaloids.

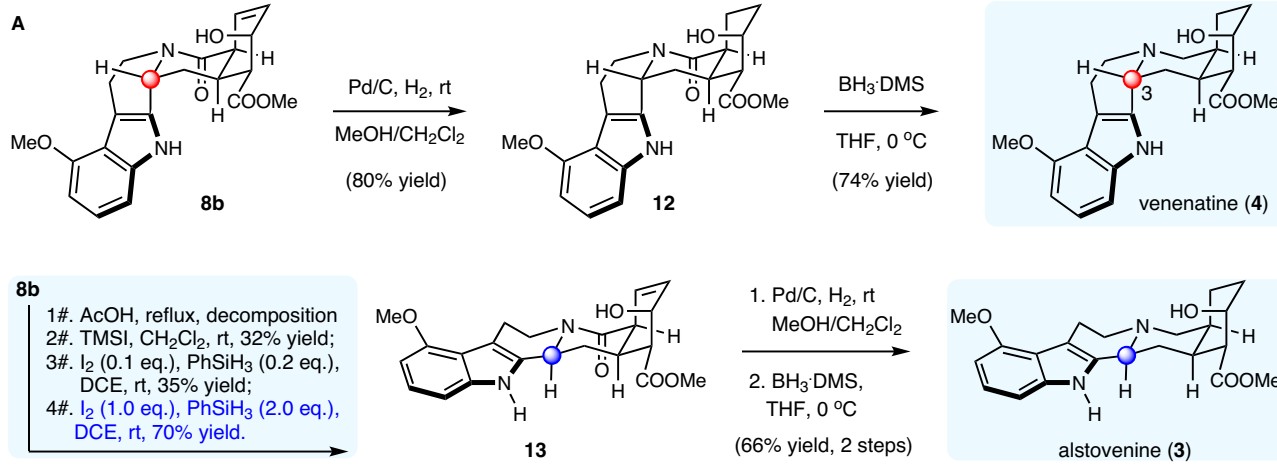

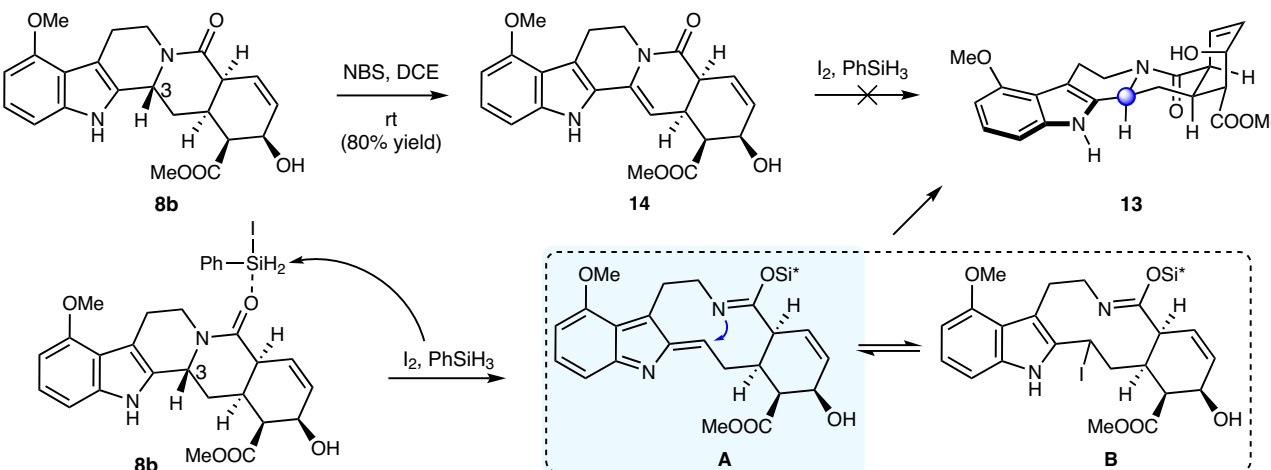

**Fig. 3 | Total synthesis of alstovenine and venenatine. A**: synthetic route to alstovenine and venenatine. **B**: control experiment and proposed mechanism for the C3-epimerization.

## Total synthesis of alstovenine and venenatine

Next, we turned to the total synthesis of the C3-epimeric yohimbine alkaloids alstovenine (**3**) and venenatine (**4**; Fig. 3A). These natural products have only been previously synthesized divergently by the group of Sarpong in racemic form in 12 steps/each[20]. From **8b**, hydrogenation of the alkene produced **12**, which underwent amide reduction using borane dimethyl sulfide (BH₃.DMS) to complete the total synthesis of venenatine (**4**) in total 5 steps from 2-pyrone **9** and tryptamine **5b**.

The synthesis of alstovenine (**3**) required the C3-epimerization of **8b**. Toward this goal, two mechanistic rationales would be feasible: (1) the oxidation of the benzylic amine portion to iminium (or its tautomer enamine) followed by kinetic axial reduction; (2) the acid mediated direct epimerization involving C3-N bond cleavage/recyclization based on the thermodynamic preference. We turned to the later approach due to its direct manner. Acetic acid (AcOH) was previously used[8,31] for the epimerization of C3 in the context of yohimbine alkaloid synthesis, however, with substrate **8b**, it only led to a complex mixture. Among several Lewis acids tested, trimethylsilyl iodide (TMSI) facilitated the conversion of **8b** to **13** in 32% yield, accompanying by several unknown side products. This result promoted us to examine other Si-I type reagents, and finally led to the identification of I₂/PhSiH₃ as a suitable reaction condition for the direct C3-epimerization. While the reagents could mechanistically be used catalytic, stoichiometric

amount of I₂/PhSiH₃ gave synthetic useful yield, presumably due to the presence of the free hydroxy group and its interaction with the reactive species. Regarding the mechanism, the oxidation of the benzylic amine portion to iminium (or its tautomer enamine) by I₂ followed by reduction with PhSiH₃ could be ruled out by the failed conversion of **14** to **13** under identical conditions (Fig. 3B). Based on previous literature studies of I₂/PhSiH₃[32,33], tentative reaction pathway was proposed (Fig. 3B). The reaction of I₂ and PhSiH₃ was reported to generate PhSiH₂I, which might serve as the Lewis acidic species for the C3-epimerization. C3-N Bond cleavage would afford intermediate **A**, which might be transiently trapped by iodide to furnish intermediate **B**. The reversible transformations between **A** and **B** might finally lead to the formation of **13** as the thermodynamically more stable product upon ring closure. Beneficial from this newly developed protocol for direct C3-epimerization, alstovenine (**3**) was made in additional two steps from **13** involving of alkene hydrogenation and amide reduction.

## Total synthesis of yohimbine, β-yohimbine and pseudoyohimbine

Finally, we turned to the chemical synthesis of yohimbine (**1a**), β-yohimbine (**1b**), and pseudoyohimbine (**2**), representing the other two stereoisomeric subfamilies of yohimbine alkaloids (Fig. 4). While the total syntheses of yohimbine (**1a**) have been widely pursued[16,18,34–37], synthetic studies on the synthesis of **2** have been relatively rare[8,38,39]

**Fig. 4 | Transition states C and D explained the outcomes of the kinetic reduction at C20.** Total synthesis of yohimbine, β-yohimbine and pseudoyohimbine.

even without full characterization data being reported. They are also C3-epimeric yohimbine alkaloids, but with the opposite stereochemistry at C20 compared with that of **3** and **4**. From **8a**, the first goal was to adjust the C20 stereochemistry. We surmised that alkene relocation followed by stereospecific reduction might be a feasible way, but the chemistry was later turned out to be more delicate. Alkene isomerization in the presence 1,8-diazabicyclo[5.4.0]undec-7-ene (DBU) delivered α, β-unsaturated amide **17**, which underwent stereospecific hydrogenation, however the stereochemistry at C20 was unchanged (**18**). As shown in Fig. 4, transition state **C** should lead to the formation of the desired stereochemistry at C20, but it was disfavored due to the presence of C16-ester at axial direction, posing notable steric hindrance. Thus, the epimerization of C16 was required prior hydrogenation. Gratifyingly, both the alkene relocation and C16-epimerization could be achieved by the treatment of **8a** with $K_2CO_3$ in methanol at 15 °C, affording **15** in 80% yield. Of note, the temperature had a notable effect on this transformation: at 0 °C, no C16-epimerization; at room temperature, partial decomposition observed. As expected, this subtle change rendered the alkene hydrogenation to occur with complete opposite selectivity (via **D**), furnishing **16** with desired C20-stereochemistry. From **16**, amide reduction delivered 17-*epi*-pseudoyohimbine (**19**). The stereochemistry at C17 could be further adjusted by a 2-step redox sequence involving Swern oxidation followed by reduction with lithium tri-*sec*-butylborohydride (L-selectride), ultimately producing pseudoyohimbine (**2**). The $^1H$-NMR data of **2** was in reasonable agreement with the reported one[8], but inconsistencies were observed, particularly for the COOMe (3.78 vs 3.92). We could not find the $^{13}C$-NMR data for further comparison, and finally confirmed the structure of **2** by X-ray crystallographic analysis.

To synthesize yohimbine (**1a**) and β-yohimbine (**1b**), the C3-epimerization of **16** was required. We were pleased to find that our newly developed $I_2$/PhSiH$_3$ condition was well applicable to substrate **16**, producing **20** in 88% yield, which underwent amide reduction to

afford β-yohimbine (**1b**). Following a known redox protocol for C17-stereochemistry adjustment by the group of Qin[7], yohimbine (**1a**) could be then made. Thus, the collective chemical synthesis of stereoisomeric yohimbine natural products **1-4** was achieved, featuring the controllable late-stage adjustment of the stereochemistry of C3, C16, C17, C20. These studies, combining with the use of bioinspired coupling for the rapid construction of the entire pentacyclic skeleton by catalytic enantioselective Pictet-Spengler/amidation cascade, cultivated a general synthetic platform for the synthesis of stereoisomeric yohimbine natural products and analogs with high degrees of stereochemical information.

## Data availability

Characterization data of intermediates and final products are provided in the Supplementary Information. The X-ray crystallographic coordinates for structures reported in this study have been deposited at the Cambridge Crystallographic Data Centre (CCDC), under deposition numbers 2287987 (**8a**) and 2287988 (**2**). These data can be obtained free of charge from The Cambridge Crystallographic Data Centre via www.ccdc.cam.ac.uk/data_request/cif. All other data are available from the corresponding author upon request.

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

## Acknowledgements

We thank the National Natural Science Foundation of China (22171161 to L.Z.), and National Key R & D Program of China (2020YFA0509300 to L.Z.) for funding support.

## Author contributions

M.T., H.L. and L.Z. designed and performed experiments and analyzed experimental data. L.Z. directed the investigations and prepared the manuscript with contributions from all authors; all authors contributed to discussions.

## Competing interests
The authors declare no competing interests.
