## [Peer Review File · Nature Communications]

Collective total synthesis of stereoisomeric yohimbine alkaloidsREVIEWER COMMENTS

Reviewer #1 (Remarks to the Author):

Tang, Lu and Zu describe the total synthesis of four yohimbine alkaloids spanning all four stereoisomeric classes. The work is inspired by the potential utility of these compounds for pharmaceutical investigation and by a “downhill stereochemical diversification” strategy. The beginning of the synthesis hinges on using 2-pyrone dimer as an alternative C10 building block to replace expensive chiral pool material. A kinetic resolution Pictet-Spengler/amidation allows for the construction of the core skeleton of the yohimbine alkaloids in an impressive three steps. In the end game, thermodynamic equilibration of the C3 stereocenter is enabled by a Lewis acid promoted ring cleavage/ring formation that leads to the requisite stereochemistry for alstovenine and β -yohimbine. Isomerization of the C16 stereocenter also allows for kinetic control in the hydrogenation of the enone to set the C20 stereocenter. Relative stereochemistry is confirmed by X-ray crystallography in several cases.

Overall, the authors report a novel approach to the yohimbine alkaloids highlighted by a thoughtful application of the pyrone dimer and careful planning of late stage stereochemical modifications. The yields are moderate to high in all cases and the SI demonstrates good support for the assigned structures. This work displays an efficiency and creativity that will be enjoyed by the synthetic community and may inspire others to apply these strategies toward other classes of natural products.

Specific comments:

Manuscript:

1. The manuscript is well organized with attractive figures. However, a number of grammatical errors detracts somewhat from the reading. A few examples are noted below, but the authors are encouraged to check the grammar carefully.
 - a. Pg. 2 Line 54: should be “arrangement of four of their five...”
 - b. Pg. 5 Line 161: “accompanying with” should be “accompanied by”
 - c. Pg 6 Lines 186–189. Consider splitting this into two sentences, as it is discussing two different things.
2. Reference 35: "Jacobsen" is misspelled
3. Some page number ranges and all reference ranges in the text use a hyphen (-) instead of an en-dash (–)
4. Temperature is given in most reaction conditions but not all (see 9 to 7, 8b to 14, 8b to 13 #1 and 2, 16 to 20). Please update this to be consistent throughout the manuscript

SI:

1. The authors report many steps on significantly smaller scale than that of subsequent downstream steps of the syntheses, e.g. the conversion of 9 to 7, 7 to 10, and 16 to 20 are reported on a significantly smaller scale than for the conversion of 10 + 5a to 8a, 8a to 15, and 20 to 1b; if scalability of any steps (especially the first two steps) is problematic, this should be acknowledged, or the transformations should be reported on larger scale
2. The ee calculated from the HPLC trace given for (+)-8a is 95%, not 96% as stated in Fig.2 and the ee calculated from the HPLC trace given for (+)-8b is 91%, not 92% as stated in Fig.2; appropriate changes should be made throughout the manuscript
3. S4-S17: This reviewer recommends including equivalents for each reagent, mass for each catalyst, mmol for each product, concentrations with respect to the substrate in the solvent,

and volumes of solutions used for washes.

4. Consider including conditions and characterization for the synthesis of the “derivative of 11a” that is used for Xray crystallography

5. It may be useful to include conditions for crystallization of all compounds which were subjected to Xray crystallography. This can help improve reproducibility and inform others working on similar molecules.

Reviewer #2 (Remarks to the Author):

Please see the following comments:

(i) I personally felt the title “downhill” stereochemical diversification is unnecessary, and the authors don’t need to “invent fancy new terms” to draw attention to their work. Essentially, this is a “stereo-divergent” synthesis which is NOT a new concept in target-oriented synthesis particularly in collective synthesis (which itself it a redundant term invented by David MacMillan in their strychno-alkaloid synthesis). At the same time, I also question the purpose of emphasizing “bioinspired coupling”, since fragment unification between tryptamine and a yohimbine “E-ring” precursor is a tactic used by virtually EVERY synthetic studies in this field. As such, it is my overall impression that the authors “over-dramatized” the title of this manuscript in order to draw audiences’ attention, which I felt entirely unnecessary and unfortunate – let the science speak for itself!

(ii) The essence of this study is the demonstration of stereochemical control, and the discovery of an unexpected stereo-induction during the fragment coupling (Pictet-Spengler reaction) in the presence of a chiral phosphoric acid catalyst. Historically, controlling the relative stereochemistry between C3 and C20 represents the thorniest challenge in the synthesis of the yohimbine family of pentacyclic alkaloids – so called the “reserpine C3 stereochemical problem”. In response, creative solutions were devised by Woodward (substrate control), Stork (reagent control), and Jacobsen (catalyst control). Furthermore, the C3-C20 anti relative stereochemical relationship is the main challenge being thermodynamically disfavored, whereas the C3-C20 syn stereochemical arrangement can be easily obtained through acid-promoted epimerization. It appears (see following comments) the authors obtained the challenging anti C3-C20 stereochemical arrangement “exclusively” during an asymmetric Pictet-Spengler fragment coupling – which sets the stage for the introduction of other stereochemical elements in the target molecules.

(iii) For the key Pictet-Spengler fragment coupling, if there is “kinetic resolution” as reported by the authors, the reported yields (70% and 80% for 8a and 8b, respectively) cannot be an accurate representation of the reaction since yield must be calculated based on compound 10 (used in 2 equivalents) and not tryptamine. Furthermore, since compound 10 is synthesized whereas tryptamine is purchased commercially, yield has to be calculated based on 10. As such, the actual yield of the asymmetric reaction, taking into enantiomeric excess of 8a and 8b, can only be ~30 to 35% - it is my impression that authors try to hide this fact to elevate the impact of their work.

(iv) Furthermore, if kinetic resolution did take place, the authors must present the optical purity analysis of the recovered compound 10. This is VERY IMPORTANT, since if both enantiomers of compound 10 reacted and the authors only isolated and reported the desired diastereoisomer, then this entire synthesis losses its value completely since the authors

simply separated the two isomers and proceeded the synthesis with one of the two diastereoisomers – i.e. no stereocontrol at all, where stereochemical control is supposed to be the highlight of the synthesis. To further convincing evidence in support of the kinetic resolution, the authors must present the “crude” NMR and HPLC analysis of the reaction mixture.

(v) Furthermore, if “kinetic resolution” is truly observed, the authors should provide theoretical and/or computation studies (energy landscape/profile, transition state structures, key non-covalent interactions, etc.) in support of the observed results and in doing so rationalize the observed absolute stereochemical outcome.

(vi) The newly discovery I₂/PhSiH₃ reagent system for the C3 epimerization is an interesting finding. However, it is puzzling that the authors reported “decomposition” under the conventional acetic acid condition. Historically, acetic acid and even other stronger acids (e.g. TFA) were successfully applied to this family of compounds, some of which contain even more elaborated functionalities. I would like the authors clarify this, together with crude NMR analysis, isolation and identification of byproducts, etc.

(vii) The late-stage stereocontrolled operations require little commentary, since there exists good literature precedence for these transformations, and the stereochemical outcome is easily predicted and dictated by the conformationally rigid pentacyclic framework.

(viii) The Supplementary Information file is by large satisfactorily prepared. However, I am puzzled by the inconsistency in reaction scales early in the synthesis. For example, compounds 7 and 10 were reported in 129 mg and 90 mg, respectively, but the Pictet-Spengler reactions (racemic and asymmetric) were performed on significantly larger scale. This should be clarified.

Reviewer #3 (Remarks to the Author):

In this manuscript, Zu and co-workers reported a general synthetic platform for the collective synthesis of four stereoisomeric yohimbine natural products with two key strategies, bioinspired coupling and downhill stereochemical diversification. Specifically, the entire pentacyclic skeleton was rapidly constructed by a catalytic enantioselective Pictet-Spengler/amidation cascade in only 3 steps, and at later stage, controllable adjustment of stereochemistry enabled the divergent synthesis of four stereoisomeric yohimbine alkaloids. In the first stage, easily accessed pyrone dimer was used to couple with tryptamine 5a or 5b to deliver the pentacyclic skeleton rac-8a. This is distinct from Scheidt’s previous synthesis, which constructed the tetracyclic ring system during the key cyclization reaction. Notably, the authors also developed an asymmetric version of the Pictet-Spengler/amidation cascade by using a chiral phosphoric acid, which led to the preparation of chiral 8a/b in excellent ee. After the pentacyclic skeleton was constructed, the resulting alkene and alpha position of N in 8a/b were utilized to orchestrate stereochemistry. In the second stage of stereochemistry adjustment, the authors taken advantage of thermodynamic or kinetic preference of substrates to convert chiral centers. Of note, during this process, the authors developed a new condition for C3-epimerization (I₂-PhSiH₃). Combination of these two stages enabled the authors to synthesize four stereoisomeric subfamilies of yohimbine alkaloids in less than 10 steps.

Overall, the synthetic strategy reported in this work is concise and efficient, and several interesting transformations, including the cascade cyclization and C3-epimerization should attract significant attention of synthetic chemists. Therefore, this manuscript is recommended for publication in Nature Communication, and some suggestions are given for the revision of this manuscript:

1. "Fig. 2. Three step construction of the pentacyclic skeleton.", "Three step" should be revised to "Three-step".
2. The format of "epi-" and "β-" should be italic "epi-" and "β-".
3. In figure 3B, if PhSiH₂I is supposed to activate oxygen of the amide, the amide moiety in structure A and B might be better drawn as the silyl imidate form (PhSiH₂ attached to the oxygen of the imidate). Apart from the possible mechanism shown in Figure 3B, are there other possible pathways? How about a retro-Friedel-Crafts reaction, followed by Friedel-Crafts cyclization? If HI is produced during the reaction, is it possible that HI catalyzes the proposed fragmentation/cyclization cascade?
4. SI: 1) the coupling constant *J* should be in italic type; 2) the title of ¹H NMR (¹H NMR (400 MHz, DMSO-d₆)) and the spectra should be better in the same page.
5. Transition state models to explain the reluctance of 11a for amidation and the selectivity of phosphoric acid catalyzed kinetic resolution should be included in Fig. 2.

Liansuo Zu, Ph. D., Associate Professor
School of Pharmaceutical Sciences
Tsinghua University, Beijing, China, 100084

86-10-62798972
zuliansuo@tsinghua.edu.cn

11, 15, 2023

Dear Editor:

Thanks for the handling of this manuscript. We appreciate the valuable comments from the reviewers as well. Below are the point-to-point response to the reviewers' comments. The changes are highlighted in yellow in the manuscript and supplementary information.

Reviewer 1

1. The manuscript is well organized with attractive figures. However, a number of grammatical errors detracts somewhat from the reading. A few examples are noted below, but the authors are encouraged to check the grammar carefully.

Response: we fixed these mentioned errors, and checked the grammar throughout the manuscript.

2. Reference 35: "Jacobsen" is misspelled

Response: we fixed that.

3. Some page number ranges and all reference ranges in the text use a hyphen (-) instead of an en-dash (–)

Response: we fixed that.

4. Temperature is given in most reaction conditions but not all (see 9 to 7, 8b to 14, 8b to 13 #1 and 2, 16 to 20). Please update this to be consistent throughout the manuscript.

Response: we updated this in the Figures.

5. The authors report many steps on significantly smaller scale than that of subsequent downstream steps of the syntheses, e.g. the conversion of 9 to 7, 7 to 10, and 16 to 20 are reported on a significantly smaller scale than for the conversion of 10 + 5a to 8a, 8a to 15, and 20 to 1b; if scalability of any steps (especially the first two steps) is problematic, this should be acknowledged, or the transformations should be reported on larger scale.

Response: we updated the scale of these mentioned transformations. The yields were in the same level, slightly lower when compared with small scale versions. 9 to 7: from 54% to 50% (5.9 g); 7 to 10: from 80% to 78% (4.9 g); 16 to 20: from 92% to 88% (679 mg). See supplementary information, pages 4, 18.

6. The ee calculated from the HPLC trace given for (+)-8a is 95%, not 96% as stated in Fig.2 and the ee calculated from the HPLC trace given for (+)-8b is 91%, not 92% as stated in Fig.2; appropriate changes should be made throughout the manuscript

Response: we changed these ees both in the main text and in the supplementary information.

7. S4-S17: This reviewer recommends including equivalents for each reagent, mass for each catalyst, mmol for each product, concentrations with respect to the substrate in the solvent, and volumes of solutions used for washes.

Response: we added the mentioned information into supplementary information.

-
8. Consider including conditions and characterization for the synthesis of the “derivative of 11a” that is used for Xray crystallography.

Response: we added the mentioned information into supplementary information, page 5.

9. It may be useful to include conditions for crystallization of all compounds which were subjected to Xray crystallography. This can help improve reproducibility and inform others working on similar molecules.

Response: we added the mentioned information into supplementary information.

Reviewer 2

1. I personally felt the title “downhill” stereochemical diversification is unnecessary, and the authors don’t need to “invent fancy new terms” to draw attention to their work. At the same time, I also question the purpose of emphasizing “bioinspired coupling”, since fragment unification between tryptamine and a yohimbine “E-ring” precursor is a tactic used by virtually EVERY synthetic studies in this field.

Response: Thank you, we agree! We changed the title to: Collective total synthesis of stereoisomeric yohimbine alkaloids. We removed these fancy terms from the title, and removed “downhill” throughout the manuscript. Regarding “bioinspired coupling”, we keep this term due to: not only the ring formation manner, but also the use of alkene for stereochemical diversification, mimic the biogenetic way.

2. For the key Pictet-Spengler fragment coupling, if there is “kinetic resolution” as reported by the authors, the reported yields (70% and 80% for 8a and 8b, respectively) cannot be an accurate representation of the reaction since yield must be calculated based on compound 10 (used in 2 equivalents) and not tryptamine.

Response: In the original submission, we explained this: “The process would be the kinetic resolution of racemic **10** in nature. While such an operation would inherently lead to the waste of half of **10**, the early-stage setting and the readily accessible of this substrate from simple 2-pyrone allowed us to use **10** in excess and use **5a** as the limiting reagent.” In the supplementary information, Table 1, we also reported the results using **10** as the limiting reagent (up to 45% yield, 95% ee); see page 6.

3. Furthermore, if kinetic resolution did take place, the authors must present the optical purity analysis of the recovered compound 10. This is VERY IMPORTANT, since if both enantiomers of compound 10 reacted and the authors only isolated and reported the desired diastereoisomer, then this entire synthesis losses its value completely since the authors simply separated the two isomers and proceeded the synthesis with one of the two diastereoisomers – i.e. no stereocontrol at all, where stereochemical control is supposed to be the highlight of the synthesis. To further convincing evidence in support of the kinetic resolution, the authors must present the “crude” NMR and HPLC analysis of the reaction mixture.

Response: we checked the ee of the recovered **10** (51% ee) in the reaction of 10 and 5a. Based on 70% yield of 8a, the observed ee close to the theoretical value (54%). The crude NMR showed the presence of unreacted 10. These results supported the kinetic resolution pathway. See supplementary information, pages 9-11.

4. if “kinetic resolution” is truly observed, the authors should provide theoretical and/or computation studies (energy landscape/profile, transition state structures, key non-covalent

interactions, etc.) in support of the observed results and in doing so rationalize the observed absolute stereochemical outcome.

Response: we added a proposed transition state into Fig. 2, which could explain the result theoretically.

5. However, it is puzzling that the authors reported “decomposition” under the conventional acetic acid condition. Historically, acetic acid and even other stronger acids (e.g. TFA) were successfully applied to this family of compounds, some of which contain even more elaborated functionalities. I would like the authors clarify this, together with crude NMR analysis, isolation and identification of byproducts, etc.

Response: we provided the crude NMR of the AcOH reaction, indicating that the reaction did lead to a complex mixture. See supplementary information, page 14.

6. The Supplementary Information file is by large satisfactorily prepared. However, I am puzzled by the inconsistency in reaction scales early in the synthesis. For example, compounds 7 and 10 were reported in 129 mg and 90 mg, respectively, but the Pictet-Spengler reactions (racemic and asymmetric) were performed on significantly larger scale. This should be clarified.

Response: Response: we updated the scale of several transformations. The yields were in the same level, slightly lower when compared with small scale versions. 9 to 7: from 54% to 50% (5.9 g); 7 to 10: from 80% to 78% (4.9 g); 16 to 20: from 92% to 88% (679 mg). See supplementary information, pages 4, 18.

Reviewer 3

1. Fig. 2. Three step construction of the pentacyclic skeleton.”, “Three step” should be revised to “Three-step”

Response: we fixed that.

2. The format of “epi-” and “β-” should be italic “epi-” and “β-”.

Response: we fixed that.

3. In figure 3B, if PhSiH₂I is supposed to activate oxygen of the amide, the amide moiety in structure A and B might be better drawn as the silyl imidate form (PhSiH₂ attached to the oxygen of the imidate). Apart from the possible mechanism shown in Figure 3B, are there other possible pathways? How about a retro-Friedel-Crafts reaction, followed by Friedel-Crafts cyclization? If HI is produced during the reaction, is it possible that HI catalyzes the proposed fragmentation/cyclization cascade?

Response: 1. we drew A and B as suggested. 2. The epimerization of C3 had been discussed in previous literature, and we examined the widely accepted two possibilities: oxidation followed by kinetic reduction & C3-N bond cleavage/recyclization. Regarding retro-Friedel-Crafts, it might be less feasible due to lacking of the driving force. 3. We checked HI, either aqueous or anhydrous, but no desired product observed.

4. SI: 1) the coupling constant J should be in italic type; 2) the title of ¹H NMR (¹H NMR (400 MHz, DMSO-d₆)) and the spectra should be better in the same page.

Response: we fixed that.

5. Transition state models to explain the reluctance of 11a for amidation and the selectivity of phosphoric acid catalyzed kinetic resolution should be included in Fig. 2

Response: we explained this as “The COOMe group played a critical role in directing the

selectivities of this transformation by dictating the facial selectivity of the Pictet-Spengler cyclization and preventing **11a** toward further amidation due to steric hindrance (Fig. 2)”. We added a transition state into Fig.2 to explain the phosphoric acid catalyzed kinetic resolution.

We are thrilled about the prospects of publishing our studies in your prestigious journal. Thank you in advance for your consideration.

REVIEWERS' COMMENTS

Reviewer #1 (Remarks to the Author):

This is a revised manuscript that reports an impressive synthesis of a series of yohimybine type natural products. The authors have done an excellent job addressing the concerns that were raised by the reviewers. I do not have more to add and I think this manuscript should be accepted.

Reviewer #3 (Remarks to the Author):

[Note from the Editor: Reviewer #3 was asked to assess also the response given to reviewer #2 who was unable to look over the revision]

The authors have adequately addressed the concerns of Reviewers 2 and 3. The manuscript should be published.

Personally, I feel like the favored transition state added in Figure 2 is not necessary.

One minor typo: line 118: amiadation should be amidation.

Liansuo Zu, Ph. D., Associate Professor
School of Pharmaceutical Sciences
Tsinghua University, Beijing, China, 100084

86-10-62798972
zuliansuo@tsinghua.edu.cn

12, 28, 2023

Dear Editor:

Thanks for the handling of this manuscript. We appreciate the valuable comments from the reviewers as well.

Reviewer 3

Personally, I feel like the favored transition state added in Figure 2 is not necessary.

One minor typo: line 118: amiadation should be amidation.

Response: We fixed the typo. Regarding the transition state in Fig. 2, this point was originally raised by the reviewer 2, and we added the transition state to explain the selectivity issue.

We are thrilled about the prospects of publishing our studies in your prestigious journal.
Thank you in advance for your consideration.